# Development of a Model Based on Delta-Radiomic Features for the Optimization of Head and Neck Squamous Cell Carcinoma Patient Treatment

**DOI:** 10.3390/medicina59061173

**Published:** 2023-06-19

**Authors:** Severina Šedienė, Ilona Kulakienė, Benas Gabrielis Urbonavičius, Erika Korobeinikova, Viktoras Rudžianskas, Paulius Algirdas Povilonis, Evelina Jaselskė, Diana Adlienė, Elona Juozaitytė

**Affiliations:** 1Department of Radiology of Lithuanian, University of Health Sciences, Eivenių g. 2, LT-50161 Kaunas, Lithuania; 2Department of Physics, Faculty of Mathematics and Natural Sciences, Kaunas University of Technology, Studentu g. 50, LT-51368 Kaunas, Lithuania; 3Oncology Institute of Lithuanian, University of Health Sciences, Eiveniu g. 2, LT-50161 Kaunas, Lithuania; 4Medical Academy of Lithuania, University of Health Sciences, A. Mickeviciaus g. 9, LT-44307 Kaunas, Lithuania

**Keywords:** FDG, head and neck squamous cell carcinoma, radiomics, PET/CT, textural analysis

## Abstract

*Background and Objectives*: To our knowledge, this is the first study that investigated the prognostic value of radiomics features extracted from not only staging 18F-fluorodeoxyglucose positron emission tomography (FDG PET/CT) images, but also post-induction chemotherapy (ICT) PET/CT images. This study aimed to construct a training model based on radiomics features obtained from PET/CT in a cohort of patients with locally advanced head and neck squamous cell carcinoma treated with ICT, to predict locoregional recurrence, development of distant metastases, and the overall survival, and to extract the most significant radiomics features, which were included in the final model. *Materials and Methods*: This retrospective study analyzed data of 55 patients. All patients underwent PET/CT at the initial staging and after ICT. Along the classical set of 13 parameters, the original 52 parameters were extracted from each PET/CT study and an additional 52 parameters were generated as a difference between radiomics parameters before and after the ICT. Five machine learning algorithms were tested. *Results*: The Random Forest algorithm demonstrated the best performance (R^2^ 0.963–0.998) in the majority of datasets. The strongest correlation in the classical dataset was between the time to disease progression and time to death (*r* = 0.89). Another strong correlation (*r* ≥ 0.8) was between higher-order texture indices GLRLM_GLNU, GLRLM_SZLGE, and GLRLM_ZLNU and standard PET parameters MTV, TLG, and SUVmax. Patients with a higher numerical expression of GLCM_ContrastVariance, extracted from the delta dataset, had a longer survival and longer time until progression (*p* = 0.001). Good correlations were observed between Discretized_SUVstd or Discretized_SUVSkewness and time until progression (*p* = 0.007). *Conclusions*: Radiomics features extracted from the delta dataset produced the most robust data. Most of the parameters had a positive impact on the prediction of the overall survival and the time until progression. The strongest single parameter was GLCM_ContrastVariance. Discretized_SUVstd or Discretized_SUVSkewness demonstrated a strong correlation with the time until progression.

## 1. Introduction

It is well known that medicine is both art and science; however, this is particularly true in the field of radiomics. Medical professionals constantly analyze images and observe shapes, colors, shades, and patterns but, nonetheless, the human eye cannot grasp all the possible features in the visual data. This makes radiomics an appealing proposition, which has the potential to identify informative combinations of features or patterns that cannot necessarily be seen with the naked eye [1]. The uncovered data can be applied in disease prevention, screening, diagnosis, staging, prognosis, evaluation of the response to therapy or radiotherapy planning, etc. [2]. Radiomics has been used for a decade; however, adopting it for clinical routine practice has been unsuccessful thus far. The main obstacles have been a lack of standardization in data acquisition, collection, and curation, followed by differences in application of machine and deep learning techniques as well as modeling issues. While some medical centers succeeded in creating a reliable training model for local use, they will still have to overcome the issues related to acceptance and trust.

Personalized cancer care for a locally advanced head and neck squamous cell carcinoma (HNSCC) is another challenge. Head and neck carcinomas are particularly unfavorable because of the complicated anatomy; presence of high biologic heterogeneity with intermixed hypoxic, necrotic, and extremely proliferative areas; limited curative treatment possibilities due to high morbidity; and subsequently, a high rate of locoregional recurrences (up to 40%) [3] and a low overall 5-year survival rate [4]. In most cases (about 90%), a local relapse and lymph node metastases occur in the first 2 years after the treatment [5]; thus, the highest priority should be given to control the locoregional disease. An effective pre-treatment response prediction might help to turn to either a patient-tailored escalation, toxicity-reducing de-escalation of chemoradiotherapy, or a switch to a different treatment option [6]. Therefore, the selection of a treatment procedure should depend on the risk of recurrence. Usual prediction factors are based on clinical data, such as stage and type of tumor, and the presence of locoregional or distant metastases. Along with the clinical features, there are several other types of prediction criteria recommended by various guidelines, including already quite well-established modalities such as 18F-fluorodeoxyglucose positron emission tomography (FDG PET/CT). FDG PET/CT has already proved its role in the staging of HNSCC and, together with several quantitative parameters, such as SUV (standardized uptake value), MTV (metabolic tumor volume), and TLG (total lesion glycolysis), may be used to select patients with a high risk of relapse [7,8,9]. Recent progress in radiomics, machine learning, and artificial intelligence technology has enabled an automated evaluation of medical images and extraction of new quantitative indices, the most notable of which has likely been the texture analysis extracted from pretreatment FDG PET/CT images [10,11].

Several radiomics studies have extracted features for analysis from FDG PET/CT performed prior to treatment. However, the literature review identified no publications that analyze the prognostic value of texture features extracted from FDG PET/CT performed after induction chemotherapy (ICT) or, more specifically, taking into account the differences between the features extracted from PET/CT prior and after the ICT. Thus, methods for estimating the possibility of recurrence based on three types of radiomic features, extracted from the pretreatment FDG PET/CT, post-neoadjuvant FDG PET/CT images, and delta parameters (differences between the pre- and the post-ICT features), while looking for the single strongest prediction factor, were explored.

The aim of the study was to construct a training model based on radiomics features obtained from FDG PET/CT in a cohort of patients with locally advanced head and neck squamous cell carcinoma (HNSCC) treated with neoadjuvant chemotherapy, to predict locoregional recurrence, development of distant metastases, and the overall survival (OS), as well as to extract the most significant radiomics features, which were included in the final model.

## 2. Materials and Methods

This study is a retrospective, single-center study. A total of 73 patients with advanced-stage unresectable head and neck cancer were referred for a pretreatment FDG PET/CT between June 2013 and June 2019. Before staging, all patients underwent physical examination, panendoscopy, and contrast-enhanced computed tomography (CECT). Staging was completed according to the American Joint Committee on Cancer (8th edition). The patients with tumors limited to the oral cavity, oropharynx, hypopharynx, and larynx were selected for further investigation. The eligibility criteria were histologically proven HNSCC of advanced stage III/IV and Eastern Cooperative Oncology Group performance status 0 or 1. The exclusion criteria were distant metastases at the initial staging and a previous history of head and neck cancer treated with chemotherapy and/or radiotherapy. Eighteen patients (18.9%) discontinued the study: one patient (1.9%) was diagnosed with a distant metastasis to the vertebral body, which led to a change in treatment tactics; twelve patients (11.3%) died after the end of the ICT treatment, without starting a radical concomitant chemoradiotherapy; and five patients (6.84%) refused further treatment after the ICT treatment or continued the treatment in a different institution. Therefore, 55 eligible patients were included in the final analysis (Table 1).

All enrolled patients received three cycles of induction chemotherapy of docetaxel cisplatin and 5-fluorouracil (DCF). The treatment was administered every three weeks. The induction chemotherapy was followed by the computer-controlled radiation therapy (CCRT).

All patients included in the study underwent two FDG PET/CT scans and CECT examinations: the first for the initial staging and the second almost two weeks after the last cycle of ICT.

Retrospective data compilations were approved by Kaunas Regional Ethics Committee for Biomedical Research (No. BE-2-41), and a written informed consent was waived for this retrospective study.

### 2.1. FDG PET/CT Acquisition

FDG PET/CT scans were obtained on the Discovery XCT (GE, USA) system with the same acquisition parameters as previously described by our group [12,13]. The patient’s preparation for PET/CT examination included fasting for more than 6 h prior to the scan with the aim to decrease the serum glucose level to 7 mmol/L. The injected activity of FDG was 4 MBq/kg of body weight. After the injection, patients remained in a quiet room for approximately 60 min.

A whole-body FDG PET/CT scan was acquired from the skull base to the mid-thighs with the patient’s arms above the head. The CT scan was obtained using a standard low-dose whole-body protocol (120 kV, 100 mA, 3.75 mm section thickness). The successive PET images were acquired using a whole-body protocol (3 min per bed position). A following localized head and neck FDG PET/CT scan was acquired with the patient positioned on a radiotherapy planning table with arms along the body. A low-dose CT scan was performed first, covering the area from shoulders to vertex (120 kV, 70 mA, and 3.75 mm). The subsequent PET images were obtained using the localized area protocol (5 min per bed position). In addition, a post-ICT head and neck scan was performed while the patient was immobilized in a treatment position with an individual thermoplastic mask (Figure 1). The attenuation correction was performed using the CT data. The images were reconstructed in a 3-dimensional mode, using the ordered subset expectation maximization algorithm (OSEM) with a 5 mm Gaussian filter on 128 × 128 and 256 × 256 matrices.

### 2.2. Determination of Regions of Interest

PET/CT images were retrieved from the medical images archiving systems and loaded into LIFEx software (version 7.3.0; www.lifexsoft.org, accessed on 22 September 2022). The primary tumor without lymph nodes on the PET images was delineated using a 3D contour around the voxel equal to or greater than 40% of SUVmax [14,15]. This computer-generated volume of interest (VOI) was rechecked by a single nuclear medicine specialist with 10 years of experience and, if needed, corrected to be similar to the manual segmentations [16]. Noise in the images was reduced by resampling FDG uptake values using 64 discrete values, boundary SUV values set to 0 and 20, and a bin width of 0.47, based on typical SUVs for HNSCC tumors [17].

Because patients were scanned twice, there were two different segmented VOIs for each patient: one before and another after the IC treatment. The original 52 quantitative parameters were extracted from each PET/CT study using the radiomics software. Features were derived from data contained in the voxels of the segmented structures and were grouped into three categories. First-order features were derived from the histogram of voxel intensities (SUVmean, SUVmax, skewness, kurtosis, etc.). Second-order textural features were based on matrices that contained information about the regional spatial arrangement of the voxels such as their homogeneity, contrast, and coarseness simulating the human perception of the image. Higher-order features such as grey-level run length features focused on local collinear voxels with the same grey level (grey-level co-occurrence matrices—GLCM, grey-level run-length matrices—GLRM, neighborhood grey-tone difference matrix wavelet decompositions—NGLDM, and grey-level size zone length matrices—GLZLM) (Table 2) [18].

### 2.3. Feature Selection

The analysis of the radiomics data was achieved in two stages. First, a correlation analysis of the acquired radiomics data was performed to assess the feasibility of using machine learning for the predictive model. Further, a selection procedure was carried out to determine the most appropriate machine learning algorithm for the radiomics case under study. Open-source tools, the original Python ML code, and R Studio for data visualization were applied.

For the correlation analysis, the relationship was chosen between selected radiomic parameters and survival parameters: progression-free survival (PFS; the period from first diagnosis to disease progression) and overall survival (OS; the period from first diagnosis to death).

*p* value < 0.05 was used as the criterion to indicate a statistically significant difference. Only parameters with statistical significance were selected for the model construction.

### 2.4. Prediction Model Selection

In order to build the most accurate prediction model for patient stratification, five commonly used machine learning (ML) algorithms were tested, comparing performance accuracy and reliability. The algorithms tested were Random Forest (RF), k-Nearest Neighbors (kNNs), Linear Discriminant Analysis (LDA), Support Vector Machines (SVMs) with a linear kernel, and Classification and Regression Trees (CARTs). These algorithms were selected to include linear (LDA), non-linear (CART, kNN), and complex non-linear (SVM, RF) methods [19].

Performance tests were applied on 4 datasets: No. 1—classical set of parameters (13 clinical parameters); No. 2—whole-body imaging radiomics parameters before chemotherapy (52 parameters), No. 3—whole-body imaging radiomics parameters after ICT (52 parameters); No. 4—whole-body delta parameters (difference between radiomics parameters before and after the ICT) (52 parameters). The classical parameter dataset was used as a starting point for testing the algorithm.

During the predictive model performance analysis process, the ML algorithms were trained and then validated using new randomly formed datasets (based on the 4 types of previously described datasets) with verification data representing 20% of the original data (11 randomly selected patients from the 55-patient dataset).

The predictive value of 5 chosen ML algorithms for the radiomics datasets was estimated using R squared (R2), the mean absolute error (MAE), and the root-mean-square error (RMSE) [19]. Groups of highly correlated ML algorithms (r > 0.8) were extracted to reduce false-positive findings. Even though many correlation values were calculated, no multiple comparisons were made and no statistical tests for r values were performed [20]. The receiver operating characteristic (ROC) curve analysis was performed to determine the optimal ML algorithm maximizing sensitivity and specificity. The area under the curves (AUC) was reported.

### 2.5. Model Construction

The following four datasets were used to predict OS and PFS: a conventional clinical, a pre-treatment PET/CT, a post-IC treatment PET/CT, and a delta PET/CT model. The conventional clinical model contained several clinical characteristics that have been linked to the survival of HNSCC patients: body mass index, age, T stage, N stage, stage according to the 8th edition of the American Joint Committee on Cancer (AJCC) guidelines, tumor location, and histology grade. The pre- and post-treatment and delta models were created by adding positive/negative findings (based on PET/CT scans) to the conventional clinical model.

The predictive models were assessed for their ability to predict OS or PFS at 3 years of follow up.

Demographic and clinical characteristics of patients, such as sex, age, clinical stage (III vs. IV), tumor site, AJCC stage, local control, progression-free survival (PFS), and overall survival (OS), were tested with the univariate analysis. Texture and gray-level parameters were tested with the multivariate analysis (Cox proportional hazard model) and then patients with the local tumor control were compared against patients with local treatment failure.

## 3. Results

### 3.1. Baseline Information

In total, the data of 55 patients, 2 (3.6%) females and 53 (96.4%) males, were analyzed in the study. Twelve patients (21.8%) were diagnosed with HNSCC stage III and 43 (78.2%) stage IV disease. The median follow-up duration of the patients was 22.26 months (range 1.32–110.33 months).

### 3.2. Radiomics-Based Models of Local Tumor Control

The classical parameter dataset (No. 1) was used as a starting point to test the algorithm. After the completion of performance tests with all five ML algorithms, it became evident that complex nonlinear methods are preferable. The Random Forest algorithm demonstrated the best performance with R2 values ranging from 0.963 to 0.998, in most datasets. R squared range results in all four datasets for Random Forest were from 0.961 to 0.996. On the other hand, the non-complex algorithms performed with more accuracy with the first (classical parameter) dataset. R2 was not as significant as expected in all other ML algorithms. For example, in the k-Nearest Neighbors algorithm, the range of R2 was between 0.88 and 0.95.

Based on the performance results, the Random Forest algorithm was identified as the most appropriate for assessing radiomics features in further analysis. A deeper analysis of 52 quantitative PET radiomics parameters in different datasets within the frames of this algorithm also revealed several further important observations.

Radiomics parameter datasets derived from post-ICT images (No. 3) and whole-body delta (No. 4) had more statistically significant parameters (each 21) compared with clinical (No. 1) and before-treatment (No. 2) image datasets (each 6, respectively). However, the whole-body delta dataset had more statistically significant parameters when compared to the other datasets.

A matrix of *p*-values was calculated for each dataset, and the data were visualized as a correlogram and a heatmap to identify any significant variables. An example of a full correlation for a set of classical parameters is presented in Figure 2.

When searching for any possible correlations in the classical parameter dataset, a significant inverse correlation was noted between the response to the IC treatment and the locoregional progression (*r* = 0.34). In contrast, a positive correlation was observed between the time to disease progression and a response to the IC treatment. The strongest correlation was detected between the time to disease progression and time to death (*r* = 0.89). A strong positive correlation was obtained between cancer stage, grade of differentiation, and the lymph node involvement (*r* = 0.69). A moderate negative correlation was obtained between body mass index before the IC treatment and relapse or positive lymph nodes (*r* = −0.38).

The next step in the analysis was to identify possible correlations between standard quantitative PET parameters and textural indices. The analysis revealed a strong positive correlation (*r* ≥ 0.8) between higher-order texture indices GLRLM_GLNU, GLRLM_SZLGE, and GLRLM_ZLNU and standard PET parameters MTV, TLG, and SUVmax. On the contrary, a strong negative correlation was detected between the standard PET parameter MTV and the higher-order texture parameter GLRLM_RLNU. Correlations between textural indices and standard PET parameters are illustrated in Figure 3.

When applying radiomics, it was important to evaluate at which point in time those features bear the most significant attributes for disease prognosis. Therefore, all three “imaging” datasets were compared through ROC (Receiver Operating Characteristic) curve analysis to determine the most suitable radiomics measurement point. ROC curves were calculated for each of the datasets without the ML model tuning and then compared. The largest area under the curve was observed when training and evaluating the ML model with radiomics parameters extracted from dataset No. 2. The ROC curve of the obtained result is provided in Figure 4.

This analysis enabled the selection of the most appropriate time point for the collection and assessment of radiomics parameters, with the correlation analysis showing that the datasets after the induction chemotherapy (No. 3) and the delta (No. 4) dataset produce the most statistically significant data.

In the delta (No. 4) dataset, one of the parameters (GLCM_ContrastVariance) was the most important for patient survival prognosis (*r* = 0.45). The GLCM_ContrastVariance parameter demonstrated a strong positive correlation—as its value increases, the patient survival increases as well. The *p*-value for this parameter was extremely low (*p* = 0.001), the lowest among all statistically significant parameters in all four datasets. The graphical distribution of survival and GLCM_ContrastVariance is shown in Figure 5.

This graph shows a group of patients with the short time of survival. Based on the other clinical parameters, those patients had a more advanced cancer stage at the time of diagnosis. On the other hand, the patients with a higher (than 41) numerical expression of GLCM_ContrastVariance demonstrated a longer survival and longer time until progression when compared to the other patients.

A relatively good correlation of 0.46 was determined between the two second-order parameters in the delta (No. 4) dataset—SHAPE_Compacity and GLRLM_HGRE (*p* = 0.001). An example of a full correlation of all 52 parameters of the delta (No. 3) dataset is shown in the heatmap (Figure 6).

Among the 52 parameters derived from dataset No. 4, a few more parameters with low *p*-value (*p* = 0.007) were noted. Statistically significant good correlations were observed between Discretized_SUVstd or Discretized_SUVSkewness and the time until progression (*r* = 0.35).

On the other hand, a negative correlation was determined between GLCM_Entropy and the chance of local progression (*r* = −0.32; *p* = 0.025), when increasing values of Discrezed_SUVmax, NGLDM_Coarseness, and GLCM_Entropy parameters led to a significant increase in the time to relapse at the treatment site (*r* = 0.33; *p* = 0.022).

The first results demonstrated that most of the features extracted from the delta dataset (No. 4) have a positive impact on the prediction of patient survival without disease and the time until death. The strongest single parameter was GLCM_ContrastVariance.

## 4. Discussion

This is most likely the first study that investigates the prognostic value of radiomics features extracted from not only staging PET/CT images, but also post-ICT PET/CT images, in a cohort of patients with locally advanced head and neck squamous cell carcinoma for prediction of the locoregional recurrence, development of distant metastases, and the overall survival.

In the study, all tumors on the PET images were segmented using a fixed relative thresholding method [14,15], rechecked by a single nuclear medicine specialist with a 10-year experience and, if needed, corrected. It is known that identifying the target lesion among other lesion-like entities in head and neck cancer patients can be challenging. The analysis of PET/CT images acquired after neoadjuvant chemotherapy is challenging due to the areas of necrosis, inflammation, etc., and the delineation of a viable tumor on post-ICT images in a group of patients responding to chemotherapy is especially complicated. In a wide variety of tumor delineation methods used so far, such as manual, thresholding-based, stochastic, learning-based, and boundary-based [21,22], not one has been proven as the most suitable tumor delineation method for PET radiomics research [19]. Therefore, the rechecking of computer-generated tumor volume by an experienced operator is necessary. All tumor volumes in this study were rechecked by a single specialist, to avoid errors caused by inter-operator variations.

The second step in the study was to test the performance of five available ML algorithms to identify the most suitable one for further analysis. Random Forest (RF), k-Nearest Neighbors (kNNs), Linear Discriminant Analysis (LDA), Support Vector Machines (SVMs) with a linear kernel, and Classification and Regression Trees (CART) were evaluated. In the study, the Random Forest algorithm outperformed others with R2 values ranging from 0.963 to 0.998 in the majority of datasets. While several previous studies reported good performance of all tested models [19,23], different algorithms performed best with specific datasets [22,23]. Machine learning conceptually aims to optimize accuracy and reproducibility as well as avoids human procedural bias in parameter selection.

Thirteen clinical parameters (classical set) and 156 parameters extracted from two PET/CT studies were used. During the training process, the ML model recalculated the collected data (from 55 patients) five thousand times to make the components of data as independent from each other as possible. As in several other studies [24], the classical parameters dataset was used as a starting point for testing the algorithm. A statistically significant correlation was detected between the time to disease progression and time to death (*r* = 0.89). A strong positive correlation was obtained between cancer stage, grade of differentiation, and the lymph node involvement (*r* = 0.69).

While the prognostic significance of standard PET quantitative parameters has been demonstrated in multiple studies, the prognostic value of textural indices has not yet been established. Several studies have analyzed different indices using various segmentation and discreditation methods. Cheng et al. [10,25] studied the prognostic value of 22 textural indices and showed an added value of a score combining TLG and two textural indices, Uniformity (Energy) and ZLNU, to predict survival. Guezennec et al. [26] reported that ZLNU was highly correlated with MTV. The analysis of our data revealed a strong positive correlation (*r* ≥ 0.8) between three textural indices GLRLM_GLNU, GLRLM_SZLGE, and GLRLM_ZLNU and standard PET parameters MTV, TLG, and SUVmax. A strong negative correlation was identified between MTV and GLRLM_RLNU. Guezennec et al. [26] reported that MTV and four textural indices extracted from the pretreatment FDG PET/CT were significant predictors of OS in the univariate analysis. In multivariate analysis, MTV and Correlation were independent prognostic factors.

When diving into the analysis of radiomics features, we aimed to determine at which point in time acquired images present the most significant prognostic potential. The results show that the radiomics features extracted from the dataset after the ICT (No. 3) and delta dataset (No. 4) achieve the best performance for the prediction of patient survival. Liu et al. analyzed the radiomics-based prediction of survival in patients with HNSCC based on pre-treatment and post-radiotherapy PET/CT [24]. They also used two sets of PET/CT images for model creations and their results showed that the post-treatment model performed significantly better than the pre-treatment and clinical models [24]. Unfortunately, when the completed treatment was unsuccessful and the radiomics-based model predicts a short patient OS, it is overdue. Accurately predicting prognosis is essential for optimizing treatment strategies in HNSCC; thus, several studies have attempted to assess the predictive potential of radiomics information from PET/CT images [6,20,24,26,27]. The primary goal of radiomics is to build clinical models using machine learning techniques, which analyze a significant number of higher-order multimodality image features and outperform specialists in visual analysis in terms of accuracy. Typically, authors select one or only a few key predictive features from their datasets for inclusion into the final model. Oh et al. [11] found that Coarseness was an independent prognostic factor of the recurrence-free survival and the overall survival. Bogowicz et al. [28] demonstrated that tumors more homogenous in CT density (decreased GLSZMsize_zone_entropy) and with a focused region of high FDG uptake (higher GLSZMSZLGE) indicate a better prognosis. Zhang et al. [29] extracted three features (minimum, skewness, and low gray level run emphasis) that were included into the combined radiomic-clinical model. In this study, the strongest single parameter was GLCM_ContrastVariance extracted from the delta (No. 4) dataset. The patients with a higher (than 41) numerical expression of GLCM_ContrastVariance demonstrated a longer survival and a longer time until progression when compared with the other patients. The other two parameters Discretized_SUVstd or Discretized_SUVSkewness demonstrated good correlation with the time until progression. On the other hand, increasing values of Discrezed_SUVmax, NGLDM_Coarseness, and GLCM_Entropy parameters led to a statistically significant increase in the time to relapse at the treatment site. Those extracted features will allow us to create a final model, which will help to perform a patient-specific clinically applicable risk stratification for HNSCC patients. A timely risk stratification enables patient-tailored escalation and de-escalation of even a change of therapy method; therefore, we recognize that early prediction is crucial for patients’ outcome.

In this study, we demonstrated that radiomics features extracted from post-ICT (No. 3) PET/CT images and the delta set (No. 4) allow for the early prediction of treatment outcome while enabling changes to patient care in case of an unfavorable prognosis. No previous studies attempted to create a radiomics-based model for early treatment outcome prediction using features extracted from delta datasets.

## 5. Conclusions

Radiomics features extracted from the delta (difference of radiomics parameters obtained from PET/CT performed before and after ICT) dataset produce the most robust information. Most of the parameters have a positive impact on the prediction of the overall patient survival and the time until progression. The strongest single parameter was GLCM_ContrastVariance. The other two parameters Discretized_SUVstd or Discretized_SUVSkewness demonstrated a strong correlation with the time until progression.

This retrospective study will be extended to create and validate the final model (the best performing radiomics features will be included), as well as to prepare it for implementation into clinical practice.

## 6. Limitations

This study was a retrospective analysis of the data collected from a pre-selected patient population; therefore, the inclusion of the patients was limited. The population included only patients with advanced stages of oropharyngeal, laryngeal, and hypopharyngeal cancers and, therefore, resulted in a dataset with a highly selected population and gender bias (53 males and 2 females), which may have had an influence on the final results.

The key limitation of this study is that further studies are required before the final model can be created. Additional studies should include a wider female population, create further training models with a combination of clinical and radiomics features, as well as investigate other possible correlations. At present, there are no other studies investigating the influence of delta values in assessing the impact of radiomic parameters on treatment planning and the management of patients; therefore, there is currently no possibility to compare the results of this study.

## Figures and Tables

**Figure 1 medicina-59-01173-f001:**
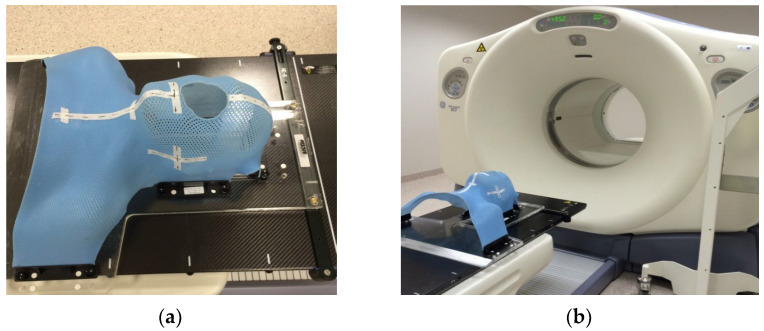
Individual thermoplastic mask (**a**) for immobilization of the patient for the post-ICT head and neck PET/CT scan (**b**).

**Figure 2 medicina-59-01173-f002:**
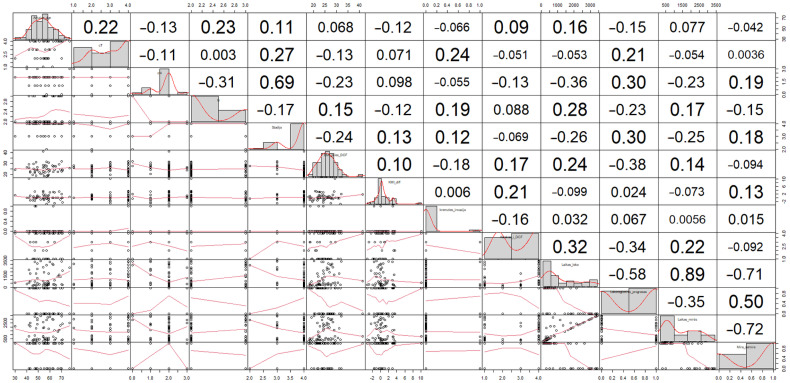
Matrix of *p*-values. Lower part of the matrix visually depicts datapoint scatter tendencies, while red trendlines depicts the dependencies of variables.

**Figure 3 medicina-59-01173-f003:**
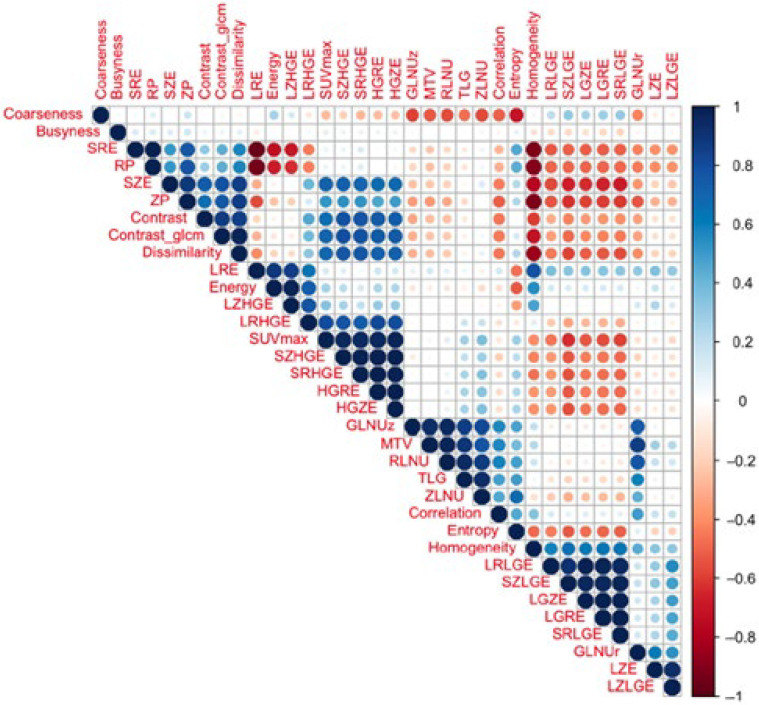
Correlation coefficient between textural indices and positron emission tomography standard parameters.

**Figure 4 medicina-59-01173-f004:**
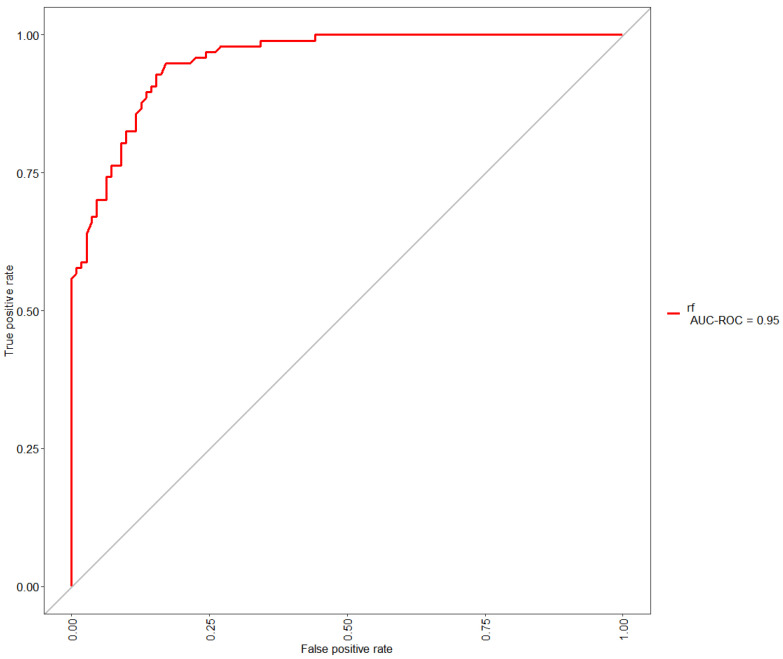
ROC curve of the Random Forest ML algorithm with whole = body imaging radiomics parameters before the ICT dataset.

**Figure 5 medicina-59-01173-f005:**
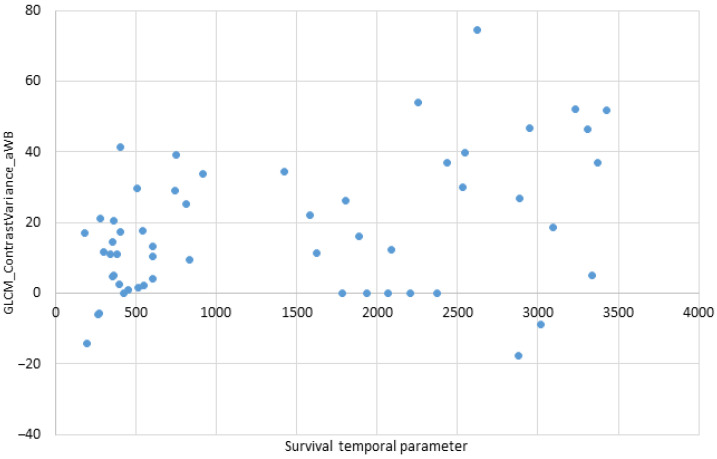
Relationship between survival and GLCM_ContrastVariance.

**Figure 6 medicina-59-01173-f006:**
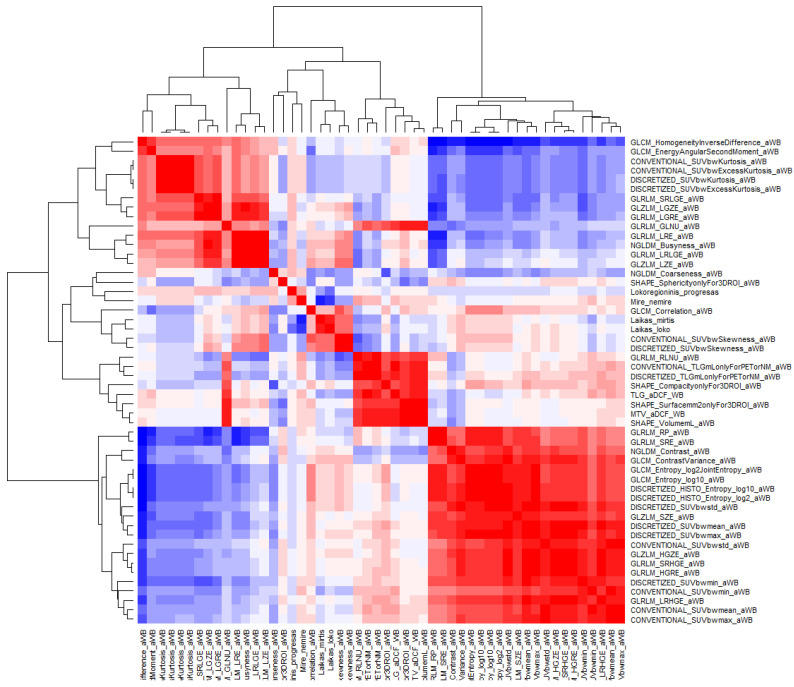
The heatmap of correlation values of second- and third-grade parameters. Red color represents complete positive correlation, while blue color—complete negative correlation.

**Table 1 medicina-59-01173-t001:** Detailed characteristics of included patients.

	Training Cohort
Number of patients	55
Number of recurrences	29
Median follow-up (months)	22.26
Age (years) ^a^	55.23 ± 9.38 (30–77)
Tumor classification	Number of patients
T1	3
T2	15
T3	13
T4	24
Nodal classification	Number of patients
N0	2
N1	10
N2	40
N3	3
Tumor location	Number of patients
Oropharynx	31
Hypopharynx	20
Larynx	2
Other	2

^a^ Minimum and maximum age.

**Table 2 medicina-59-01173-t002:** Detailed PET/CT and delta-radiomic features extracted from PET images of HNSCC patients.

First-order features	PET parameters	SUVmin, SUVmean, SUVstd, SUVmax, SUV_Skewness, SUV_Kurtosis, SUV_Excess Kurtosis, TLG, MTV, Discretised SUVmin, Discretised SUVmean, Discretised SUVstd, Discretised SUVmax, Discretised_Skewness, Discretised_Kurtosis
Second-order features	Intensity features	HISTO_Skewness, HISTO_Kurtosis, HISTO_Entropy_log10, HISTO_Entropy_log2, Discretised_HISTO_Entropy_log10, Discretised_HISTO_Entropy_log2
	Shape features	SHAPE_Volume_ml, SHAPE_Volume_vx, SHAPE_Sphericity, SHAPE_Compacity
	GLCM ^a^	GLCM_Homogenicity, GLCM_Energy, GLCM_Contrast, GLCM_Correlation, GLCM_Entropy_log10, GLCM_Entropy_log2
Third-order features	GLRLM ^b^	GLRLM_SRE, GLRLM_LRE, GLRLM_LGRE, GLRLM_SRLGE, GLRLM_SRHGE, GLRLM_LRLGR, GLRLM_LEHGE, GLRLM_GLNU, GLRLM_RLNU, GLRLM_RP
	NGLDM ^c^	NGLDM_Coarseness, NGLDM_Contrast, NGLDM_Busyness
	GLZLM ^d^	GLZLM_SZE, GLZLM_LZL, GLZLM_LGZE, GLZLM_HGZE

^a^—Grey level co-occurrence matrix. ^b^—Grey Level Run Length Matrix. ^c^—Neighborhood gray-level dependence matrix. ^d^—Gray-level zone length matrix.

## Data Availability

Data sharing is not applicable to this article.

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
