# Peer review of "Development of a Model Based on Delta-Radiomic Features for the Optimization of Head and Neck Squamous Cell Carcinoma Patient Treatment"

_medicina, 2023, doi:10.3390/medicina59061173_

Round 1
Reviewer 1 Report
In the article "Development of a radiomics based model for the optimization of head and neck squamous cell carcinoma patient treatment" Šedienė et al. I propose a radiomic model to predict the prognosis of patients with locally advanced head and neck cancers with the help of models based on the delta variation of radiomic features extracted before and after DCF protocol induction chemotherapy. To the radiomic featres extracted from PET/CT pre treatment and post induction chemotherapy, the authors add specific PET/CT parameters for the development of models based on 5 machine learning algorithms. The study also summarizes the characteristics of the patients and radiomic features in 2 tables. I would recommend that in table 2 there is an organization in such a way that there are no "empty boxes". Being a table dedicated to features and PET/CT characteristics, I would not call it "characteristics of patients" but rather "detailed PET/CT and delta-radiomic features". One figure out of the 6 is dedicated to immobilizing the patient in order to make a PET/CT quality acquisition, the others to the radiomic algorithm. The results are current and deserve to be published. However, I would mention in the title that it is a delta-radiomics study, the number of articles exploiting this concept being smaller and the topic more interesting than simple "radiomics". For the part of creating the model and implementing the machine learning algorithm, I would recommend the evaluation of the article by an expert in informatics/AI
Author Response
Response to Reviewer 1 Comments
Point 1: I would recommend that in table 2 there is an organization in such a way that there are no "empty boxes". Being a table dedicated to features and PET/CT characteristics, I would not call it "characteristics of patients" but rather "detailed PET/CT and delta-radiomic features".
Response 1: Table 2 was reorganized; the title was changed into “Detailed PET/CT and delta-radiomic features extracted from PET images of HNSCC patients”.
Point 2: I would mention in the title that it is a delta-radiomics study, the number of articles exploiting this concept being smaller and the topic more interesting than simple "radiomics".
Response 2: “Development of a radiomics based model for the optimization of head and neck squamous cell carcinoma patient treatment” was changed to “Development of a model based on delta-radiomic features for the optimization of head and neck squamous cell carcinoma patient treatment”.

Reviewer 2 Report
The authors report the results from a retrospective study where radiomics is used to prognosis in patients with head and neck cancer.
A lot of different models and variables are presented, but the selection process in which variables or models are in- or excluded is not sufficiently described. I.e. table 2 (which has the same table legend as table 1?) a lot of parameters are mentioned - where there any selection process behind the selection of these parameters? Suggest revising the whole methods section to a more concise presentation of the selection of parameters/models, where there any thoughts beforehand on which variables to include – or randomly were variable randomly included if the were a significant correlation?
Further the table (Table 2) has a poor layout - consider to do a more simplistic layout and reverse the order of columns and lines. Finally the abbreviations are not explained GLCM?, GLRLM?
In order to assess the generality of the study - how many records were screened and excluded from the study?
Although head and neck cancer is 2-4 fold more common in men than women, the population is very skewed with less than 4% female - have the authors any suggestion to why there is such a skewed gender distribution and the possible effects on the generality of the study?
Figures 2, 3 and 6 are very difficult to read due to the small font and heavy information load - suggest revising and possibly, to address these figures in a supplement in a larger size.
Figure 5: What does "Survival parameters" stand for? Is it survival in days? Please revise.
How did the radioomics models preform compared to standard known risk factors - both simple variables as gender, age, cancer stage, lymph node involvement but also FDG-variables as SUV measures, TLG, MTV? Could be presented in the form of ROC-curves. In short, does the radiomics add significantly to standard practice?
Please revise the conclusion - take in to account that 1) it is a retrospective study, 2) that correlation do not mean causation, and at best the results are hypothesis generating. Please avoid the formulation "most statistical significant data".
Some of the points above could also be included in a Limitation section at the end of the discussion.
Author Response
Response to Reviewer 2 Comments
Point 1: A lot of different models and variables are presented, but the selection process in which variables or models are in- or excluded is not sufficiently described. I.e., table 2 (which has the same table legend as table 1?) a lot of parameters are mentioned - where there any selection process behind the selection of these parameters? Suggest revising the whole methods section to a more concise presentation of the selection of parameters/models, where there any thoughts beforehand on which variables to include – or randomly were variable randomly included if there were a significant correlation?
Response 1: The whole methods section was reviewed and reorganized; now we hope the presentation of parameters is more understandable.
Point 2: Further the table (Table 2) has a poor layout - consider doing a more simplistic layout and reverse the order of columns and lines.
Response 2: Table 2 was reorganized; the title was changed into “Detailed PET/CT and delta-radiomic features extracted from PET images of HNSCC patients”.
Point 3: Finally, the abbreviations are not explained GLCM, GLRLM?
Response 3: The abbreviations are explained in the text and under the Table 2.
Point 4: In order to assess the generality of the study - how many records were screened and excluded from the study?
Response 4: Information on patient selection was added to “The materials and methods” section: 73 patients were included into the study in the beginning, eighteen patients discontinued the study for various reasons, and therefore, 55 eligible patients were included into the final analysis.
Point 5: Although head and neck cancer is 2-4 fold more common in men than women, the population is very skewed with less than 4% female - have the authors any suggestion to why there is such a skewed gender distribution and the possible effects on the generality of the study?
Response 5: The reason for the skewed population might be explained by several factors. Firstly, the two main primary tumor sites are hypopharyngeal and oropharyngeal cancers (92,7%). According to statistics, there is a 3-5-fold higher risk of oropharyngeal cancer and 7-10-fold higher risk of hypopharyngeal cancer in men than in women. (Chiruvella V, Guddati AK. Analysis of Race and Gender Disparities in Mortality Trends from Patients Diagnosed with Nasopharyngeal, Oropharyngeal and Hypopharyngeal Cancer from 2000 to 2017. Int J Gen Med. 2021 Oct 2; 14:6315-6323; Park JO, Nam IC, Kim CS, Park SJ, Lee DH, Kim HB, Han KD, Joo YH. Sex Differences in the Prevalence of Head and Neck Cancers: A 10-Year Follow-Up Study of 10 Million Healthy People. Cancers (Basel). 2022 May 20; 14(10):2521.).
Secondly, women tend to be diagnosed at earlier stages as they seek medical attention as soon as they notice the first symptoms. In our study only patients with advanced stage III/IV were included. (Dittberner A, Friedl B, Wittig A, Buentzel J, Kaftan H, Boeger D, Mueller AH, Schultze-Mosgau S, Schlattmann P, Ernst T, Guntinas-Lichius O. Gender Disparities in Epidemiology, Treatment, and Outcome for Head and Neck Cancer in Germany: A Population-Based Long-Term Analysis from 1996 to 2016 of the Thuringian Cancer Registry. Cancers (Basel). 2020 Nov 18;12(11):3418).
Due to the two reasons mentioned above, the hypopharyngeal and oropharyngeal cancer patients in our hospital are mainly men, so the representation of the gender in our study corresponds to the real situation in the clinic. To add more women to the study we would need a much larger cohort of patients, however, as induction chemotherapy is not so widely used and women cohort with locally advanced disease is rather small, this would be a difficult task to achieve. Nevertheless, according to the results of published studies, outcomes of locally advanced oropharyngeal and hypopharyngeal cancer treatment are similar in men and women, therefore, results of our study, which are achieved from mainly men cohort are reliable and applicable for women as well.
Point 6: Figures 2, 3 and 6 are very difficult to read due to the small font and heavy information load - suggest revising and possibly, to address these figures in a supplement in a larger size.
Response 6: Figures 2, 3, 4, 5 and 6 were enlarged. The Journal should decide if Figure 6 should be placed as an annex.
Point 7: Figure 5: What does "Survival parameters" stand for? Is it survival in days? Please revise.
Response 7: Figure 2 title was rechanged, survival parameters changed in to days.
Point 8: How did the radiomics models perform compared to standard known risk factors - both simple variables as gender, age, cancer stage, lymph node involvement but also FDG-variables as SUV measures, TLG, MTV? Could be presented in the form of ROC-curves. In short, does the radiomics add significantly to standard practice?
Response 8: Since main goals of predictive models was to find possible correlations with radiomic parameters and then attempt to predict PFS (the period from first diagnosis to disease progression), overall survival (OS; the period from first diagnosis to death), other parameter evaluation is outside of the scope of this study and this additional data would not improve it.
We are creating the predictive models and testing them to find out if those models add significantly to standard practice, the process in itself is the means of progression and movement towards dedications of more routine functions of a physician to artificial intelligence. So maybe currently it does not add significantly to standard care, but it will in the very near future.
Point 9: Please revise the conclusion - take into account that 1) it is a retrospective study, 2) that correlation do not mean causation, and at best the results are hypothesis generating. Please avoid the formulation "most statistical significant data".
Response 9: The conclusion revised. Also, we have corrected the text, trying to avoid the term “statistically significant data”.
Point 10: Some of the points above could also be included in a Limitation section at the end of the discussion.
Response 10: The Limitation section is added at the end of the discussion.

Round 2
Reviewer 2 Report
A few comments:
In the abstract conclusion - please avoid the term "most statistically significant data".
In the limitation section - the retrospective nature of the study, potential gender bias and potential bias in the highly selected population should be mentioned.
Author Response
Response Round 2 to Reviewer 2 Comments
Point 1: In the abstract conclusion - please avoid the term "most statistically significant data
Response 1: Changed into “most robust data”.
Point 2: In the limitation section - the retrospective nature of the study, potential gender bias and potential bias in the highly selected population should be mentioned.
Response 2: The limitation section was rewritten.
This study was a retrospective analysis of the data collected from a pre-selected patient polpulation, therfeore, the inclusion of the patients was limitted.
The population included only patients with advanced stages of oropharyngeal, laringeal and hypopharyngeal cancers and, therefore, resulted in a dataset with a highly selected population and gender bias (53 males and 2 females), which may have had an influence on the final results.
The key limitation of this study is that further studies are required before the final model can be created. Additional studies should include a wider female population, create further training models with a combination of clinical and radiomics features as well as investigate other possible correlations.
At present, there are no other studies investigating the influence of delta values in assessing the impact of radiomic parameters on treatment planning and management of patients, therefore, currently there is no possibility to compare the results of this study.